# An SEM Model of Learning Engagement and Basic Mathematical Competencies Based on Experiential Learning

**Lu Sun and Longhai Xiao ***

School of Education, Zhejiang University, Hangzhou 310058, China
* Correspondence: longhaixiao2022@163.com

**Abstract:** Primary school mathematics is one of the most important subjects in primary school learning, and basic mathematical competencies are an important component of the response to academic achievement. Improving students' basic competence in mathematics is one of the important goals of teaching mathematics in primary schools. Research has shown that experiential learning has an impact on basic competencies in mathematics, attitudes toward mathematics, and self-efficacy in mathematics. Therefore, this study explores the structural model that fits the relationship between experiential learning and basic competencies in mathematics using a linear model. This study uses a sample of 263 primary school students to explore the influential relationships between learning engagement, mathematical attitudes, mathematical self-efficacy, and basic mathematical competencies after experiential learning. The study revealed that the model had a good fit, with learning engagement, mathematical attitudes, and mathematical self-efficacy all having significant effects on basic mathematical competencies; in addition, behavioral engagement had insignificant effects on mathematical attitudes and mathematical self-efficacy. This study can infer through one year of experiential learning and based on the structural model developed that experiential learning in mathematics can increase students' learning engagement in mathematics learning and positively influence mathematical attitudes and mathematical self-efficacy, thus positively influencing students' performance in basic mathematical competencies.

**Keywords:** primary mathematics; experiential learning; basic mathematics skills; SEM model; mathematical attitudes; mathematical self-efficacy

## 1. Introduction

Elementary school mathematics is one of the most important subjects in elementary school learning, and basic mathematics skills are an important component of the response to academic achievement. In recent years, more and more researchers have begun to look at the impact of various forms of curriculum reform on psychological factors, and then to explore the impact of these psychological factors on basic mathematics skills. The elementary school level is a critical period for the development and shaping of mathematical learning abilities and habits, and it is important to explore the mechanisms by which mathematical learning patterns affect basic mathematical abilities and improve various psychological factors. Therefore, this study aims to investigate the relationship between the learning engagement with mathematics and mathematics basic competence after experiential learning in mathematics; whether mathematical attitudes and mathematical self-efficacy play a mediating role in experiential learning; and to make some suggestions for subsequent model exploration and improvement.

With the introduction of China's double reduction policy, how to improve the traditional teaching model, for example, using experiential learning in the classroom to develop students' positive mathematical attitudes, enhancing mathematical self-efficacy, and improving basic mathematical competencies, is now the primary issue in education. Research has shown that experiential learning has an impact on basic mathematical competence,

mathematical attitudes, and mathematical self-efficacy. When teachers evaluate authentic learning tasks, they need to use evaluation techniques, and in this case, the evaluator's approach to this evaluation is crucial [1–3]. Therefore, valid and reliable student learning engagement scales, attitude scales, and self-efficacy scales will help in the identification of such contexts and the implementation of instructional programs, as well as in responding to the direct effects of student learning. The direct effects of student learning can be reflected through the testing of Basic mathematical competencies.

Therefore, this study will examine the analysis of students' learning engagement, mathematical attitudes, mathematical self-efficacy, and basic mathematical competence after the experiential learning of elementary school students as a sample, and establish the appropriate pathways and model structures to provide an analytical model reference for whether conducting experiential learning in elementary school mathematics in the future can positively affect students' basic mathematical competence, as well as to predict the relationship and impact of experiential learning and basic competencies in mathematics.

## 2. Review of the Literature

### 2.1. Definition of Experiential Learning in Mathematics

Experiential learning in mathematics is a student-centered approach to learning mathematics in which students acquire new mathematical knowledge, skills, and attitudes through a combination of practice and reflection, based on their existing knowledge and experiences. In mathematics, experiential learning can promote elementary school students' understanding of the meaning of numbers and calculations, etc., stimulate students' interest in exploring the wonders of elementary school mathematics, and effectively guide them to develop a scientific way of thinking, while at the same time, it can increase students' motivation to learn mathematics and exercise their independent learning skills, and the communication between teachers, students, and peers is promoted in the process of operational practice, prompting communication and interaction between teachers and students [4].

The new Chinese mathematics curriculum standards require that in a learning situation, the concepts, ideas, and tools of a discipline are used to integrate mental processes and manipulative skills to solve problems in real situations, reflecting the process of "problem situation—modeling—solution verification"; the teaching approach should clarify the knowledge and skills learned, and use various activities to teach and design. This is in line with the concept of experiential learning in mathematics, which requires students to formulate hypotheses, predict results, choose methods, and express their understanding clearly, through communication and cooperation.

Experiential learning emphasizes that learning should create certain life situations that are centered around facilitating the development of practical skills, allowing students to have rich experiences, and allowing students to learn to express their own experiences and know how to understand people's expressions of their experiences, thus seeking to provide an environment that engages students in social practices, engages them in inquiry-based learning, and supports them in establishing a positive identity [5]. Problem solving, hands-on learning by doing, and deep inquiry activities are all compatible with the concept of experiential learning, so integrating mathematics classrooms with experiential learning can be used to explore the role of experiential learning in facilitating mathematics learning for elementary school students.

Students in the early elementary grades are still at the stage of intuitive thinking and practical experience building. Therefore, the learning of computational and statistical content in the mathematics classroom in the early grades is approached in a way that is related to the experiential learning approach to mathematics, with an eye on students' behavioral engagement, cognitive engagement, and affective engagement, and through physical and mental engagement in concrete contextual practices to reflect on learning, summarize experiences, and build knowledge systems [6].

## 2.2. Definition of Basic Competencies in Mathematics

The basic mathematical ability includes four basic abilities: arithmetic ability, logical thinking ability, spatial imagination ability, and problem-solving ability [7–10]. Arithmetic ability refers to the ability to complete various operations quickly, correctly and reasonably; logical thinking ability refers to the ability to think correctly using the forms, laws and methods of logical thinking; spatial imagination ability is the ability of people to observe, analyze and think abstractly about the spatial form of things; problem-solving ability is the ability to use the mathematical knowledge or methods learned to apply to real life and solve practical problems. These basic abilities are formed and developed gradually with the accumulation of life experience and continuous learning, they have different levels at the same time or at different times, and are closely related to each other, interlocking and interpenetrating, forming a whole. Research shows that childhood is an important period when human mathematical abilities begin to develop, a critical period for mastering mathematical concepts, performing abstract operations, and beginning to form comprehensive mathematical abilities [11–15].

Basic mathematical competencies are not innate; the family environment and learning environment in which an individual lives plays an important role in his or her development, and the individual's own psychology and physiology also play an active and dynamic role in development. Childhood is a critical period for the formation of good basic mathematical competencies. According to the Soviet psychologist Kruchetsky, basic mathematical ability should include nine components: the ability to do arithmetic, logical reasoning, the ability to think in simple terms, the ability to reverse mental processes, the ability to think flexibly, numerical memory, and spatial concepts, among which the ability to generalize is the core of mathematical ability [15].

Chinese psychologist Lin Chongde, on the other hand, believes that the basic mathematical ability is a multifaceted, multilevel, multiform, and multiconnected system consisting of three abilities, arithmetic ability, logical thinking ability, and spatial imagination ability, and five qualities of thinking, agility, flexibility, profundity, originality, and criticality. The main factors that influence the development of students' basic mathematical competencies are mainly biological and socialization factors [16]. Among the socialization factors, schooling factors play an important role. Additionally, among the schooling factors, whether the educational and learning styles are appropriate or reasonable for the students is included, and among all schooling factors, the two main ones are teachers and teaching [17], such as experiential learning, which has a greater impact on the development of students' basic competencies in mathematics.

## 3. Research Frameworks

### 3.1. Purpose of the Study

In this study, by sorting out the relationship between learning styles and learning attitudes, learning styles, and self-efficacy, we focus on the effects of experiential learning in mathematics on mathematical attitudes and mathematical self-efficacy on basic mathematical competencies [18]. By investigating students' behavioral engagement, cognitive engagement, and affective engagement as well as mathematical attitudes and mathematical self-efficacy after experiential learning in mathematics, we find out whether there is a significant effect between each factor and basic mathematical competence.

### 3.2. Experiential Learning in Mathematics

Dewey emphasized that learning is the process of creating knowledge through the transformation of experience [19]; David Kolb's proposed Experiential Learning Theory (ELT) emphasizes the role of experience in constructing knowledge [20]. By integrating David Cooper's learning cycle into teaching and learning activities, the study found that experiential learning helped students achieve their learning goals while providing them with irreplaceable exposure [21]. David Cooper's learning cycle to introduce some skills in critical reading instruction [22].

Recent research has shown that teachers in learning with strong content knowledge play a key role in student achievement [23]. Primary school mathematics teachers are the ones who shape the mathematical knowledge and behavior of most students later in life. Experiential learning is one implementable way to teach and learn [24]. Some studies have shown that from kindergarten onwards, students' reading and mathematics achievements can be significantly influenced by the quality of classroom teachers and, most importantly, these influences do not dissipate over time but they remain strong predictors of future academic achievement [25]. Most school teachers are increasingly concerned with student achievement in mathematical performance, mathematical skills, and attitudes toward mathematics. To address these concerns, teachers then need to decide how to improve their mathematics, for example, by modifying the content and skills emphasized in the curriculum, changing or supplementing instructional materials, changing content approaches, and changing the use of teaching and learning methods [26].

Due to their age, primary school students are curious and eager to learn. Research has shown that in primary school mathematics, it is particularly important for students to improve their attitudes and self-efficacy in mathematics by allowing them to truly learn through 'experience' and by making effective and positive assessments of their learning.

### 3.3. Attitudes to Mathematics

Research shows that teachers can develop positive attitudes toward learning, even when there is little family support as a starting point; teachers need to get students to see mathematics themselves as challenging, interesting, and useful. Attitudes create a self-perpetual cycle: children with positive beliefs about mathematics do well, which makes them enjoy mathematics and feel good about themselves, and students with negative beliefs fall further behind, which reinforces their low expectations and feelings [27].

Mathematical attitudes have multiple important effects on students' mathematical achievements. On the one hand, mathematical attitudes positively influence students' mathematical achievements in the present; if students hold positive mathematical attitudes, they are likely to enjoy the process of learning mathematics and put more effort into it, resulting in good mathematical achievement [28,29], which is important for students' success [30]; whereas students with negative mathematical attitudes may avoid or postpone the task or even develop negative emotions, such as aversion [31,32]. On the other hand, mathematical attitudes change as experience and knowledge are acquired [33], which in turn, affects the subsequent mathematical achievement [34,35]. For example, a meta-analysis by Ma and Kishor (1997) showed that many students have relatively positive attitudes toward mathematics in their early education, yet their attitudes toward mathematics become increasingly negative as they remain in education, thus hindering their mathematical achievement. Although mathematical attitudes have an important impact on mathematical achievement, there has long been more focus on external factors such as teaching methods and content, and the relative neglect of the development of students' mathematical attitudes.

There is also research that shows that attitudes are extremely important in problem solving [36]. Positive attitudes can improve an individual's academic ability, while negative attitudes are associated with a low motivation to learn [33]. Mathematical attitudes play a special and critical role in the process of teaching and learning mathematics [34]. As defined by Ma and Kishor, mathematical attitudes include the liking or disliking of mathematics, the tendency to engage in or avoid mathematical activities, the belief that one is good or bad at mathematics, and the belief that mathematics is useful or useless [35]. Attitudes toward mathematics have a significant impact on students' mathematic achievements, and several studies have shown a positive relationship between attitudes toward mathematics and mathematics achievement [37]. If students have positive attitudes toward mathematics, they may enjoy the process of learning mathematics and put in more effort to achieve good grades in mathematics [29], which is important for students' academic [30]. Moreover, even though mathematical attitudes can influence mathematics performance, the specific mechanisms by which mathematical attitudes influence mathematics performance

have rarely been studied and the development of students' mathematical attitudes has been neglected. Therefore, it is necessary to discuss in detail the mechanisms by which mathematical attitudes affect mathematics achievement, so as to improve the importance that parents and teachers attach to students' mathematical attitudes and work together to help students maintain the best mathematical learning attitudes and work together to help students maintain the best mathematical learning attitudes and improve their mathematical achievements. Park and Taekyung (2022) showed that according to the experiential learning approach, teachers can adopt diverse teaching methods, such as learning by doing, simulation, and participation activities [38]. Ramírez, María-José and Allison, Pete (2023) showed that "enjoyable experiences" can positively influence attitudes over time [39]. Yvette also found through her research that students prefer experiential learning that is facilitative (creative thinking) and realistic (transferable to the real world) [40].

### 3.4. Mathematical Self-Efficacy

In education, many studies have confirmed the tremendous impact of self-efficacy on student academics. Self-efficacy on students' academic motivation [41], academic achievements [42], and other influences such as learning strategies [43] and perseverance in the face of academic setbacks [44]. In light of this strong evidence, it is necessary for educational researchers and practitioners to explore the mechanisms underlying the development of self-efficacy. Several researchers have conducted in-depth studies on the relationship between mathematical self-efficacy and students' mathematics achievement. Mathematics achievement in this context is defined as students' test scores or their performance in the mathematics courses they take. Their findings suggest that mathematical self-efficacy is a better predictor of achievement than mathematical anxiety, mathematical self-concept, mental ability, prior mathematical knowledge, and perceived utility of mathematics [45–47]. Additionally, mathematical self-efficacy predicted mathematics performance better than intelligence test scores and personality traits (agreeableness, self-awareness, and emotional instability). Self-efficacy also influences learners' self-awareness, emotional instability, extraversion (openness), and self-esteem [48]. There is a consensus that mathematical self-efficacy is strongly and positively related to student achievement in mathematics [49,50]. That is, high mathematical self-efficacy is associated with high achievement in mathematics, while low mathematical self-efficacy is associated with low mathematics achievement. Mathematical self-efficacy in mathematics learning was demonstrated by Zakariya using an innovative structural equation model with an instrumental variables approach to demonstrate a causal relationship between students' mathematical self-efficacy in mathematical tasks and students' mathematical achievement [51]. It is worth mentioning that improving mathematical self-efficacy is central as an important way to improve students' mathematical achievements. As previous research has shown, mathematical self-efficacy not only predicts student performance in mathematics, but it also has a potential causal relationship with achievement [51–54]. The pedagogical implications of this relationship are that mathematics teachers have the opportunity to enhance student performance in mathematics by strengthening mathematical self-efficacy and therefore, to improve student academic performance in mathematics. For a student, a student with a high sense of mathematical self-efficacy reduces his mathematical anxiety and thus his apprehension about his mathematical performance, thus reducing the risk of failure in mathematics [51,55]. On the other hand, these associations between mathematical self-efficacy and final learning outcomes, such as the association between a student's mathematical performance and the risk of failure in mathematics, give greater credence to the idea that interventions to improve students' self-efficacy contribute to the meaningfulness of interventions that improve students' sense of efficacy.

Various studies have shown that self-efficacy tends to influence human behavior, as people tend to choose what they believe is within their reach. People with a stronger sense of self-efficacy are more confident that their efforts will lead to success and therefore put in more effort; people with an emphasized sense of self-efficacy tend to be more optimistic

when they encounter difficulties and are therefore less likely to give up. Mathematical self-efficacy refers to the degree to which an individual is confident that he or she can complete tasks in mathematical situations and mathematical problems [56]. Studies have shown that mathematical self-efficacy is significantly and positively related to mathematics achievement, and that mathematical self-efficacy has a positive effect on mathematic ability [57].

Mathematical self-efficacy positively predicts mathematics achievement [58–60]; students' positive learning significantly influences math performance [61], and more confident students tend to exhibit higher self-efficacy and are more likely to report higher levels of mathematics performance.

## 4. Research Methodology

### 4.1. Initial Model Construction

This study is based on the structural model and measurement model of SEM (Structural Equation Modeling), using Amos24.0 for data analysis, model construction, and validation. Based on the previous literature to explore and collate, measurement models are built and analyzed by Amos24.0 in order to find the most appropriate model to fit the sample data. The impact of experiential learning in primary school mathematics on attitudes toward mathematics, mathematical self-efficacy, and the impact of primary school mathematics experiences on basic math skills through the impact on attitudes toward mathematics and mathematical self-efficacy is explored. A conceptual diagram of the model for this study is shown in Figure 1.

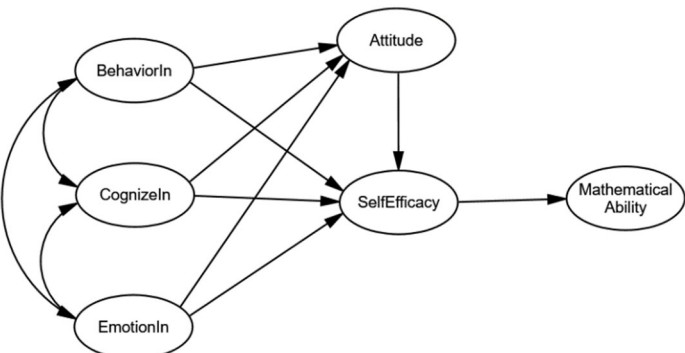

**Figure 1.** Conceptual diagram of the model.

### 4.2. Research Hypothesis

A review of the literature shows that experiential learning in mathematics is useful for the improvement of basic mathematical competencies. Basic mathematical competencies are divided into numeracy, spatial reasoning, and problem-solving skills. Basic mathematical competencies include numeracy, spatial, and reasoning competencies, which are measurable through specific measurements. Based on the characteristics of basic mathematical competencies, this study uses the structure and measurement models of SEM to explore the structural model of optimal fit between mathematical attitudes, mathematical self-efficacy, and basic mathematical competencies after implementing experiential learning in mathematics.

Chin (1998) states that SEM models should be analyzed without making declarative assumptions about each pathway, but rather by assessing the fit of the overall SEM model to the sample [60]. Therefore, the first hypothesis of this study is that the expectation model and the covariance matrix of experiential learning in mathematics do not differ from the sample covariance matrix: $S - \sum (\Theta) = 0$, S is the sample covariance matrix and $\sum (\Theta)$ is the expectation model covariance matrix of experiential learning in primary mathematics, thus forming the following research hypothesis.

Research hypothesis:

**Hypothesis 1 (H$_0$).** *The expectation model for experiential learning in mathematics does not differ from the covariance matrix to the sample covariance matrix.*

**Hypothesis 1 (H$_{01}$).** *The basic mathematical ability model and covariance matrix do not differ from the sample covariance matrix.*

**Hypothesis 1 (H$_{02}$).** *The mathematical attitude model and mathematical self-efficacy model did not differ from the covariance matrix and the sample covariance matrix.*

Mathematical self-efficacy has been considered critical in terms of its impact on improving basic mathematical competencies. As a result, the following hypotheses were developed:

**Hypothesis 2 (H$_1$).** *There is no difference in the impact of mathematical self-efficacy on basic mathematical ability.*

As this study also explores the psychological dimension, the impact of experiential learning in mathematics on mathematical self-efficacy through attitudes toward mathematics, the following hypothesis was formed:

**Hypothesis 3 (H$_2$).** *There is a partial mediating effect of mathematical attitudes on mathematical self-efficacy.*

### 4.3. Research Tools

This questionnaire contains five sections: 1. sample statistical variables, 2. Learning Engagement Questionnaire, 3. mathematical self-efficacy, 4. attitudes toward mathematics, and 5. basic mathematics skills.

Sample statistical variables included the following: gender, grade level, and whether or not they participated in after-school services. A 5-point Likert scale was used for all of this study. The operational definitions for this study were mainly drawn from the following: the Learning Participation Questionnaire (Cronbach's α of 0.82) developed by Kong Qi Ping (2003) [61], the Primary Mathematics Learning Efficacy Scale (Cronbach's α of 0.83) developed by Liu Dianzhi (2003) [62], Tapiahe Marsh (2004) [63], adapted and revised by Lin and Huang (2014) [64], the Attitude Toward Mathematics Questionnaire (ATMI) (Cronbach's α of 0.92), and the Heidelberg University Basic Competence Test in Primary Mathematics Scale in Germany, finalized by J. Haffner and K. Baron, adapted by Dr. Li Li (2005), and revised the Chinese Basic Competence Scale in Primary Mathematics (Cronbach's α of 0.85), in line with Chinese primary school students [65].

### 4.4. Sample Study Estimation

SEM is a large sample analysis technique and SEM models require a ratio of observed variables to a sample size of between 1:10 and 1:15 [66], so a sample size of between 200 and 400 is more appropriate. In this study, the revised and improved method of Mac-Callum, Browne, and Sugawara (1996) was used and was programmed in R to calculate the sample size using RMSEA [66]. In this case, H0 was set to 0.05 and the check power was 0.08 in the RMSEA calculation; Hypothesis 1 was set to 0.06 in the RMSEA calculation for the opposing hypothesis and the resulting sample size requirement ranged from 89 to 125 for the condition of 525 degrees of freedom. The effective sample size for this study is 263, which meets the sample size requirement for SEM analysis.

### 4.5. Method of Sampling the Study Sample

Primary school students in a public primary school were used as the target population for this study. A total of 270 questionnaires were distributed, and 263 valid questionnaires were returned after deducting those with incomplete answers. There were 146 boys and 117 girls in this questionnaire. There were 67 students in Grade 2, 134 in Grade 3, and 62 in

Grade 6 in the questionnaire. The grades covered all stages of primary school, including lower, middle, and upper primary school, and were consistent with the target population of the Basic Competency Scale in Mathematics study and were representative of each school level.

The subjects of this survey were from different grades in different schools in Ningbo, Zhejiang Province, China, but all were classes that participated in experiential learning in elementary school mathematics and were taught by teachers of the researcher's experiential pilot group. In the surveyed sample, the number of male and female students was similar; the number of those who participated in after-school services was 223, and the number of those who did not participate in after-school services was 40, with the majority of those participating in after-school services.

*4.6. Measurement and Structural Pattern Analysis*

4.6.1. Validation of Convergent Validity

Validated factor analysis (CFA) is an important part of SEM analysis, and Thompson notes that researchers need to analyze the measurement model before analyzing the structural model for SEM analysis, which correctly reflects the study's confirmation of factors. The reduction in the factors of the CFA variables measured in this study was modified according to the second-order model proposed by Kline (2005) [67], where the measurement model was examined before the structural model was analyzed, and if the fitness of the modified measurement model was acceptable, then the second step, the full SEM model evaluation, was carried out. In this study, CFA analysis was conducted for all the constructs. The four constructs of the model are engagement in learning after experiential learning in mathematics, which includes behavioral engagement, cognitive engagement, and affective engagement; attitude toward mathematics, which includes self-confidence, mathematical values, mathematical enjoyment, and mathematical motivation; self-efficacy in mathematics, which includes goal confidence, course competence, and course responsiveness; and basic competence in mathematics, which includes mathematical operations, spatial logic, and overall mathematics score.

After CFA analysis of the four constructs of experiential learning in mathematics, attitudes toward mathematics, self-efficacy in mathematics, and basic competence in mathematics, the factor loadings for all the constructs ranged from 0.64 to 0.87 and all the *p*-values were significant; the component reliability of the constructs ranged from 0.77 to 0.90, respectively, and the convergent validity of the constructs ranged from 0.53 to 0.75 (see Table 1).

The CFA analysis of all the constructs revealed that the factor loadings were greater than 0.5; the compositional reliability was greater than 0.6; the convergent validity was greater than 0.5; and the square of the multivariate correlation coefficient was greater than 0.5. In this study's model, all the constructs met the criteria, except for mathematical operations and spatial logic, which were slightly below 0.5 in the basic mathematical ability but still within the acceptable range; therefore, these four constructs have convergent validity [51,68].

4.6.2. Validation of Differential Validity

Discriminant validity is a test of whether two different constructs are statistically different from each other in terms of correlation. In this study, the AVE values were estimated using Amos 24.0 and the Pearson correlation coefficients between each construct were calculated after the AVE values were rooted, and if the standardized correlation coefficients for each construct were greater than the rest of the constructs, then there was differential validity between the constructs. Table 2 shows that the correlation coefficient of the mathematical self-efficacy construct was 0.865, which was greater than the correlation coefficients of the other constructs, so there was discriminant validity between the mathematical self-efficacy construct and the other constructs; the correlation coefficient of the mathematics attitude construct was 0.773, which was greater than the correlation coefficients of the

other constructs, so there was discriminant validity between the mathematics attitude construct and the other constructs; and the correlation coefficient of the mathematics basic competence construct was 0.901, which was greater than the correlation coefficients of the other constructs. The correlation coefficients of the basic competence construct were greater than those of the other constructs, so the basic competence construct was found to have differential validity of the other constructs; the correlation coefficient of the experiential learning construct was 0.758, which was greater than those of the other three constructs, so the basic competence construct was found to have differential validity with the other constructs.

**Table 1.** Confidence analysis of potential conformations.

| Structure | Title | Significance Estimates of Parameters | | | | Factors Load Capacity | Question Reliability | Component Reliability | Convergent Validity |
| --- | --- | --- | --- | --- | --- | --- | --- | --- | --- |
| | | Unstd. | S.E. | t. Value | $p$ | Std. | SMC | CR | AVE |
| Experiential learning in mathematics | Behavioral engagement | 1.000 | | | | 0.772 | 0.596 | 0.801 | 0.574 |
| | Cognitive engagement | 0.871 | 0.089 | 9.787 | *** | 0.690 | 0.476 | | |
| | Emotional engagement | 1.052 | 0.104 | 10.127 | *** | 0.806 | 0.650 | | |
| Basic mathematical competencies | Mathematical operations | 1.000 | | | | 0.668 | 0.446 | 0.773 | 0.536 |
| | Spatial logic | 1.062 | 0.124 | 8.581 | *** | 0.641 | 0.411 | | |
| | Total competence scores | 1.832 | 0.221 | 8.275 | *** | 0.867 | 0.752 | | |
| Mathematical self-efficacy | Target confidence | 1.000 | | | | 0.869 | 0.755 | 0.899 | 0.748 |
| | Course competence | 1.213 | 0.070 | 17.251 | *** | 0.870 | 0.757 | | |
| | Course response | 1.135 | 0.067 | 16.982 | *** | 0.856 | 0.733 | | |
| Attitude to mathematics | Sees mathematics as fun | 1.000 | | | | 0.732 | 0.536 | 0.873 | 0.633 |
| | Mathematical motivation | 1.198 | 0.092 | 13.041 | *** | 0.868 | 0.753 | | |
| | Confidence in mathematics | 1.270 | 0.097 | 13.081 | *** | 0.862 | 0.743 | | |
| | Mathematical values | 0.757 | 0.082 | 9.251 | *** | 0. 708 | 0.501 | | |

Note: * $p < 0.05$, ** $p < 0.01$, *** $p < 0.001$. (Same as below).

**Table 2.** Construct differential validity analysis.

| | AVE | Mathematical Self-Efficacy | Attitude to Mathematics | Basic Mathematical Competencies | Experiential Learning in Mathematics |
| --- | --- | --- | --- | --- | --- |
| Mathematical self-efficacy | 0.748 | **0.865** | | | |
| Attitude to mathematics | 0.598 | 0.769 | **0.773** | | |
| Basic mathematical competencies | 0.812 | 0.247 | 0.140 | **0.901** | |
| Experiential learning in mathematics | 0.574 | 0.758 | 0.741 | 0.152 | **0.758** |

### 4.6.3. Overall Suitability of the Model

When validating SEM theoretical models, a good model fit is a necessary condition for SEM analysis, with a better fit representing a closer match between the model matrix and the sample matrix. Several indicators of overall model fit were evaluated in this study, including χ2 check, ratio of χ2 to degrees of freedom, fit indicator (GFI), adjusted fit indicator (AGFI), root means squared error of approximation (RMSEA), non-standard fit indicator (NNFI), asymptotic fit indicator (IFI), comparative fit indicator (CFI), and standardized square root (SRMR). The model indicators in this study are referenced from Schreiber, McDonald, Boomsma, Stephenson, Hoyle and Panter, and Schreiber Stage et al. [55,69–73]. In addition, three more indicators are needed in the model fitness, namely the Akira Pool Information

Indicator (AIC), the Bayesian Information Indicator (BIC), and the Expected Cross Validity Indicator (ECVI). The model structure of this study is illustrated in Figure 2, and the model fitness is shown in Table 3.

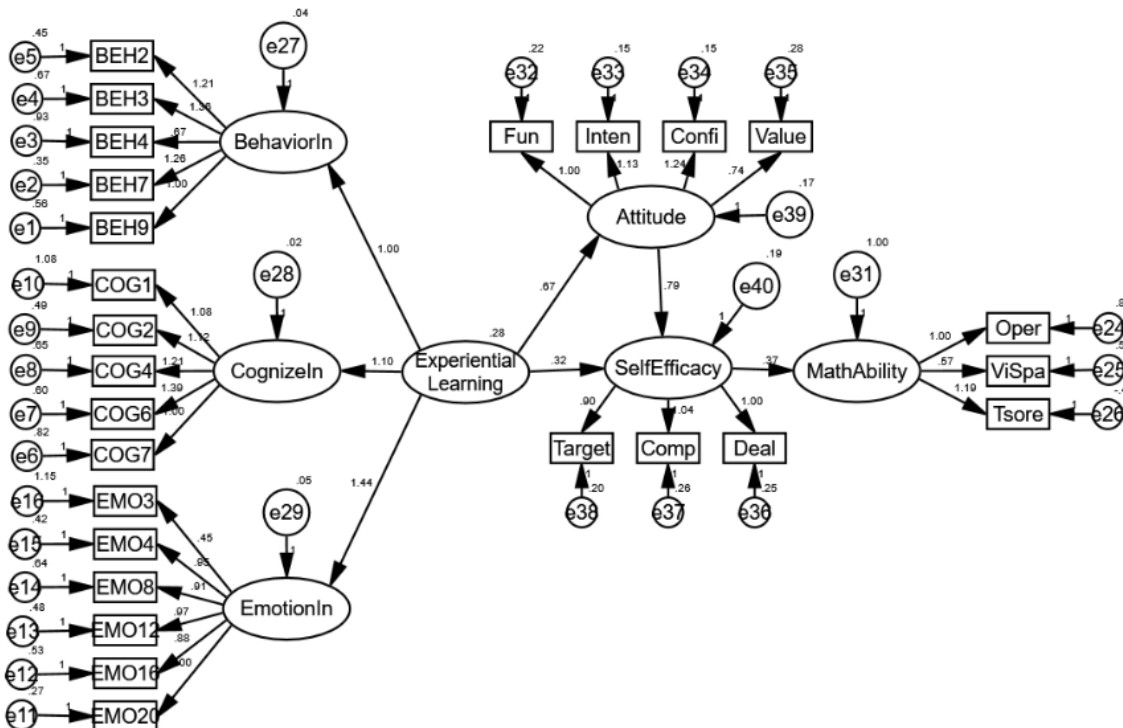

**Figure 2.** Statistical model of the structure of expectations for experiential learning in mathematics.

**Table 3.** Overall model fit of the desired model.

| Adaptation Indicators | Ideal Requirement Criteria | Models |
| --- | --- | --- |
| χ2 | The smaller the better | 515.808 |
| DF (degrees of freedom) | The bigger the better | 61 |
| Normed Chi-square (χ2/DF) | 1 < χ2/DF < 3 | 1.124 |
| GFI | >0.9 | 0.971 |
| AGFI | >0.9 | 0.943 |
| RMSEA | <0.08 | 0.022 |
| TLI (NNFI) | >0.9 | 0.996 |
| CFI | >0.9 | 0.997 |
| IFI | >0.9 | 0.997 |
| Hocltcr's N (CN) | >200 | 234.596 |
| ECVI | The smaller the better | 0.489 |
| AIC | The smaller the better | 128.58 |
| BIC | The smaller the better | 235.745 |

As can be seen from Table 3, for this study model, the smaller the values of the fitness indicators ECVI, AIC, BIC, and χ2, the better, and the greater the degrees of freedom the better. The GFI, AGFI, NNFI, CFI, and IFI of the model fitness indicators are all greater than 0.9 and the CN is greater than 200. The RMSEA is in the 95% confidence interval, it does not contain 0.08, and it is less than 0.08, indicating that the RMSEA of less than 0.08 is not a coincidence; therefore, the study model has good fitting.

### 4.6.4. Intermediary Effects

A mediating effect is an approximate significance test for the indirect effect of the independent variable on the dependent variable through a mediator [74]. A system of

equations can be specified to connect X to Y through multiple mediators [75] and few showed that mediating effects can be computed through Amos to calculate standard error values and point estimates, leading to a check Z value [76]. One inference technique for mediated effects is the coefficient product method, which is known as the Sobel Test. Although the Sobel Test has some uses as a complement to the Baron and Kenny method, the Sobel Test has a major drawback in that it requires the sampling distribution of indirect effects to be normal, but in actual sampling surveys, the sample usually presents a non-sincere distribution. In recent years, Fairchild, Mac Kinnon, Taborga, and Taylo have introduced a measure of effect size in which the proportion of variance in the response variable is explained by the indirect effect, but this indicator also has the property of influencing the variance in a response variable that has no proportion, so its value can also be negative.

In this study, standard error values and point estimates were estimated by executing the Bootstrapping procedure with 1000 replicate samples at a 95% confidence interval set at the time of execution. As can be seen from Table 4, the point estimate for experiential learning through mathematics on mathematical attitudes was 0.529, with a standard error of 0.105, and a calculated check Z value of 5.038, which meets the standard value of $Z > 1.96$, indicating that there is an indirect effect of experiential learning on mathematical attitudes; the point estimate for attitudes through mathematics on mathematical self was 0.315, with a standard error of 0.114, and a calculated mediated. The effect check Z value is 2.763, which meets the criterion $Z > 1.96$, indicating that there is a direct effect of experiential learning on attitudes toward mathematics; the point estimate of experiential learning through mathematics on mathematical self-efficacy is 0.844, with a standard error of 0.130, and the mediating effect check Z value is calculated to be 6.492, which meets the criterion $Z > 1.96$, indicating that there is a total effect of experiential learning on mathematical There is a total effect on self-efficacy.

**Table 4.** Mediating effects of the structural model.

| SIE | Point Estimate | Product of Coefficients | | Bias Corrected 95% CI | | Percentile 95% CI | |
|---|---|---|---|---|---|---|---|
| | | SE | Z | Lower | Upper | Lower | Upper |
| Indirect effects Experiential learning in mathematics → mathematical self-efficacy | 0.529 | 0.105 | 5.038 | 0.365 | 0.788 | 0.365 | 0.788 |
| Direct effect Experiential learning in mathematics → mathematical self-efficacy | 0.315 | 0.114 | 2.763 | 0.109 | 0.551 | 0.109 | 0.551 |
| Total effect Experiential learning in mathematics → mathematical self-efficacy | 0.844 | 0.130 | 6.492 | 0.627 | 1.138 | 0.627 | 1.138 |

*4.7. Model Path Coefficients*

The analysis of the path coefficients of this model shows that the regression coefficients of behavioral engagement on attitudes toward mathematics and behavioral engagement on self-efficacy in mathematics are not significant, and the remaining constructs show significant effects between the two (See Table 5).

*4.8. Model Cross Validity*

To further validate the stability of the model, this study verified the invariance of the two cohorts by gender, the two cohorts with or without participation in after-school services, and the three cohorts by grade level, respectively. This includes measuring the

factor loadings, structural path coefficients, and factor covariances of the model, and if they do not differ from each other, the model can be said to have considerable stability, i.e., cross validity. In this study, using SPSS 26.0, the sample was divided into the following groups: 1. boys and girls; 2. those who participated in after-school services and those who did not; and 3. grade 2, grade 3, and grade 6. The analysis was then conducted separately for the different cohorts using Amos 24.0.

**Table 5.** Path coefficients in the structural model.

| Structure | Std. | Non-Std. | S.E. | C.R. | *p* | SMC |
|---|---|---|---|---|---|---|
| Behavioral engagement → Attitude to mathematics | 0.238 | 0.090 | 0.047 | 1.909 | 0.056 | |
| Cognitive engagement → Attitude to mathematics | 0.841 | 0.303 | 0.048 | 6.336 | *** | 0.318 |
| Emotional engagement → Attitude to mathematics | 0.414 | 0.152 | 0.048 | 3.200 | 0.001 | |
| Attitude to mathematics → Mathematical self-efficacy | 0.570 | 0.719 | 0.104 | 6.888 | *** | |
| Behavioral engagement → Mathematical self-efficacy | 0.140 | 0.067 | 0.056 | 1.189 | 0.234 | |
| Cognitive engagement → Mathematical self-efficacy | 0.309 | 0.140 | 0.061 | 2.314 | 0.021 | 0.525 |
| Emotional engagement → Mathematical self-efficacy | 0.249 | 0.115 | 0.058 | 1.997 | 0.046 | |
| Mathematical self-efficacy → Basic mathematical competencies | 0.586 | 0.368 | 0.093 | 3.962 | *** | 0.049 |

Note: * $p < 0.05$, ** $p < 0.01$, *** $p < 0.001$. (Same as below).

As can be seen from Table 6, under the assumption that the structural model is correct, the two groups of boys and girls were compared. 1. The factor loadings were set as equal between the two groups, and the model structure had a total of 26 factor loadings that were set as equal (DF = 26), with a cardinality (CMIN) increase of 27.082, and the *p*-value of the test result was 0.405, which did not reach the significance level ($p < 2$. In addition to maintaining the limitations of the measurement model, three additional structural path coefficients were set, and the chi-squared value (CMIN) increased by 2.695, with a *p*-value of 0.000, reaching a significance level of $p < 0.05$, indicating that the three structural path coefficients were not equal. The coefficients are not all equal, but the ΔCFI is 0.000, which does not reach the standard indicator of 0.05 and meets the criterion of ΔCFI ≤ 0.05, so these three conformational path coefficients are in the acceptable range. 2. In addition to maintaining the structural model path coefficients, the settings of the variance and covariance matrices are increased by one more, and the chi-squared value (CMIN) increased by 0.498, with a check result of $p = 0.000$, reaching a significance level of $p < 0.05$, but the ΔCFI is 0.000, not reaching the standard indicator of 0.05, and meeting the criterion of ΔCFI ≤ 0.05, indicating that it is acceptable for this one variance and covariance to be set equal, so this one variance and covariance are equal.

**Table 6.** Comparison of structural models without deformation for different gender cohorts.

| Model | DF | CMIN | *p* | NFI | IFI | RFI | TLI | CFI | ΔCFI |
|---|---|---|---|---|---|---|---|---|---|
| | | | | Delta 1 | Delta 2 | rho1 | rho2 | | |
| Measurement weights | 26 | 27.082 | 0.405 | 0.006 | 0.007 | −0.003 | −0.004 | 0.908 | −0.001 |
| Structural weights | 3 | 1.695 | 0.638 | 0.000 | 0.000 | −0.001 | −0.001 | 0.908 | 0.000 |
| Structural covariances | 1 | 0.498 | 0.480 | 0.000 | 0.000 | 0.000 | 0.000 | 0.908 | 0.000 |
| Structural residuals | 6 | 8.068 | 0.233 | 0.002 | 0.002 | 0.000 | 0.000 | 0.908 | 0.000 |
| Measurement residuals | 26 | 53.961 | 0.001 | 0.012 | 0.014 | 0.003 | 0.003 | 0.901 | −0.007 |

As can be seen from Table 7, under the assumption that the structural model is correct, there is a comparison of the two clusters of participants in after-school services with those not participating in after-school services. 1. Setting the factor loadings equal between the two clusters, the model structure has a total of 20 factor loadings that are set as equal (DF = 20) and the chi-squared value (CMIN) increases by 23.519, with a check result *p*-value of 0.264, which does not reach the significant level. In addition to maintaining the limitations of the measurement model, six additional structural path coefficients were set

and the chi-squared value (CMIN) increased by 2.258, with a *p*-value of 0.894, which did not reach the significance level (*p* < 0.05), indicating that these six structural path coefficients are all equal. 2. In addition to maintaining the structural model path coefficients, the number of variances and covariance matrix increased by one setting, and the chi-squared value (CMIN) increased by 0.382, with a check result *p* of 0.536, which did not reach the significance level (*p* < 0.05), indicating that these one variance and covariance are all equal.

**Table 7.** Comparison of whether structural models are involved in after-school service clusters without deformation.

| Model | DF | CMIN | *p* | NFI | IFI | RFI | TLI | CFI | ΔCFI |
| | | | | Delta 1 | Delta 2 | rho1 | rho2 | | |
|---|---|---|---|---|---|---|---|---|---|
| Measurement weights | 20 | 23.519 | 0.264 | 0.005 | 0.006 | −0.003 | −0.003 | 0.893 | −0.001 |
| Structural weights | 6 | 2.258 | 0.894 | 0.000 | 0.001 | −0.002 | −0.002 | 0.894 | 0.001 |
| Structural covariances | 1 | 0.382 | 0.536 | 0.000 | 0.000 | 0.000 | 0.000 | 0.894 | 0.000 |
| Structural residuals | 6 | 4.619 | 0.594 | 0.001 | 0.001 | −0.001 | −0.001 | 0.894 | 0.000 |
| Measurement residuals | 26 | 47.400 | 0.006 | 0.010 | 0.012 | 0.001 | 0.001 | 0.889 | −0.005 |

As can be seen from Table 8, under the assumption that the structural model is correct, the three clusters of Year 2, Year 3, and Year 6 were compared. 1. The factor loadings were set as equal between the two clusters, and the model structure had a total of 40 factor loadings that were set equal (DF = 40), with an increase in cardinality (CMIN) of 33.589, with a *p*-value of 0.753, which did not reach a significant level. 2. In addition to maintaining the limitations of the measurement model, adding 12 more structural path coefficients to the settings increased the chi-squared value (CMIN) by 32.240 and the *p*-value for the test result was 0.001, reaching a significance level of *p* < 0.05, indicating that the 12 structural path coefficients were not equal. The structural path coefficients are not all equal, but the ΔCFI is 0.000, which does not reach the standard indicator of 0.05 and meets the criterion of ΔCFI ≤ 0.05, so these 12 conformational path coefficients are in the acceptable range. 3. In addition to maintaining the structural model path coefficients, the settings of the two variance and covariance matrices are increased by 2 more, the chi-squared value (CMIN) increased by 5.493, and the check result *p* was 0.064, which did not reach the significance level of *p* < 0.05, indicating that the 2 variances and covariances were all equal.

**Table 8.** Comparison of structural models without deformation for different grade clusters.

| Model | DF | CMIN | *p* | NFI | IFI | RFI | TLI | CFI | ΔCFI |
| | | | | Delta 1 | Delta 2 | rho1 | rho2 | | |
|---|---|---|---|---|---|---|---|---|---|
| Measurement weights | 40 | 33.589 | 0.753 | 0.007 | 0.008 | −0.006 | −0.007 | 0.892 | 0.002 |
| Structural weights | 12 | 32.240 | 0.001 | 0.007 | 0.008 | 0.003 | 0.004 | 0.867 | −0.025 |
| Structural covariances | 2 | 5.493 | 0.064 | 0.001 | 0.001 | 0.000 | 0.001 | 0.866 | −0.001 |
| Structural residuals | 12 | 27.090 | 0.008 | 0.006 | 0.007 | 0.002 | 0.002 | 0.862 | −0.004 |
| Measurement residuals | 52 | 101.304 | 0.000 | 0.021 | 0.026 | 0.004 | 0.005 | 0.849 | −0.013 |

*4.9. Model Application*

4.9.1. Research Methodology

The experimental class implemented experiential learning strategies such as project-based learning, thematic activity-based learning, mathematical activities, and experiential learning combining interdisciplinary and practical activities; the control class conducted regular classroom instruction, such as regular instruction based on the three-dimensional objectives and key points of the lesson, but did not exclude the need for any cooperative group inquiry activities. In this study, students' engagement, mathematical attitudes, mathematical self-efficacy, and mathematical process competence were pre-tested in the two classes one semester after their first year of schooling. In the following year, the

experimental class underwent a year of experiential learning based on experiential learning strategies, while the control class underwent regular traditional classroom teaching during the year. At the end of the year, post-tests were administered to the experimental class and the control class on the dimensions of engagement in learning, attitude toward mathematics, self-efficacy in mathematics, and basic competencies in mathematics, and data were collected and analyzed using SPSS 26.0 to analyze whether there were significant differences in the dimensions of engagement in learning, attitude toward mathematics, self-efficacy in mathematics, and basic competencies in mathematics after the year of experiential learning. The data were used to compare whether experiential learning had a positive impact on these dimensions.

4.9.2. Sampling Methods

The sample for this study was drawn from the two classes taught by the researcher. The two classes were evenly distributed at the time of entry and the academic achievement of the two classes differed significantly in the first month of the academic achievement test, with the control class scoring 2.3 points higher than the experimental class. The questionnaire was administered at the time of the survey and the pre-test was administered to students when they first entered the primary school for one semester, a semester in which both classes focused on familiarizing themselves with the rules of primary school learning. Additionally, the classroom was taught in a traditional classroom; the post-test was administered to the experimental class after a year of experiential learning, while the control class also had a year of traditional classroom learning. During the sampling process, 67 questionnaires were distributed for both the pre-test and post-test, and 67 questionnaires were returned. Of these, 34 were in the experimental class and 33 for the controller class (see Figure 3).

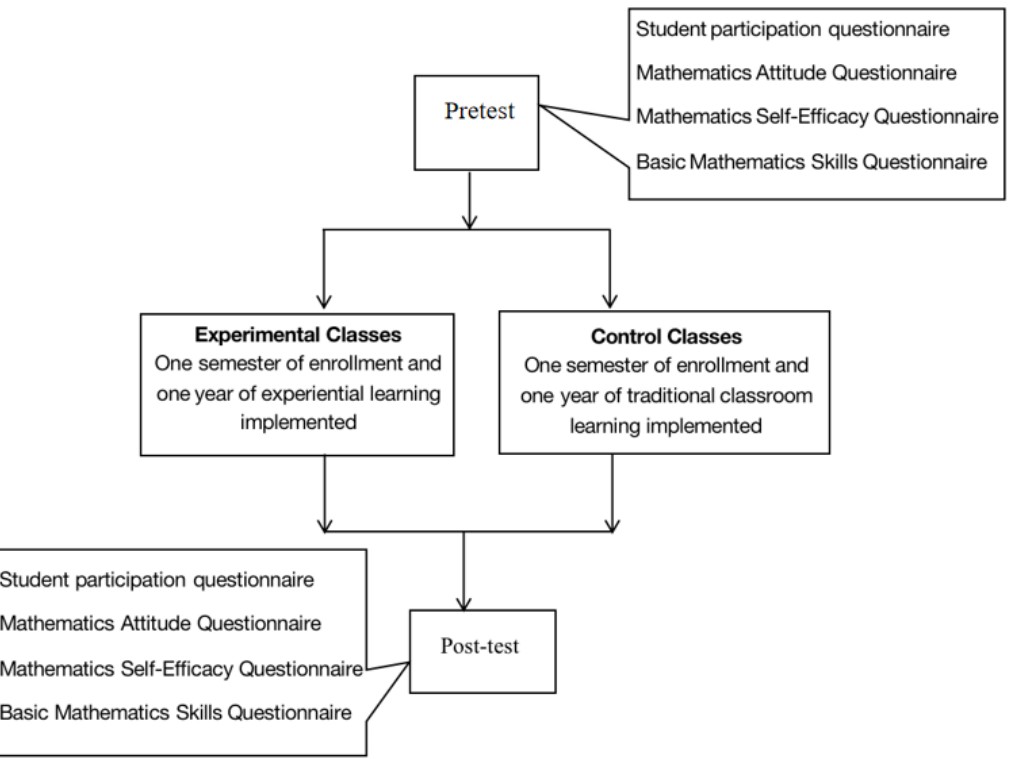

**Figure 3.** SEM model-based experimental research pathway diagram.

4.9.3. Data Analysis

The independent samples *t*-test and between-group analysis of the pre-test and post-test data collected from the experimental and control classes using SPSS 26.0 showed that

there were no significant differences between the two classes in the dimensions of student engagement, mathematical attitudes, mathematical self-efficacy, and basic mathematical competencies before the implementation of experiential learning; whereas there were significant differences between the experimental class and the control class in the dimensions of student engagement, mathematical attitudes, mathematical self-efficacy, and basic mathematical competencies after one year of experiential learning and one year of traditional classroom learning. After one year of experiential learning, there were significant differences between the two classes in the dimensions of student engagement, mathematical attitudes, mathematical self-efficacy, and basic mathematical competencies. There were no significant differences between the post-test and the pre-test in the control class. The same pattern was found in the sub-dimensions of student engagement, mathematical attitudes, mathematical self-efficacy, and basic mathematical competencies when analyzed using SPSS 26.0 (See Tables 9–11).

**Table 9.** Independent samples *t*-test for experimental and control classes.

| Testing Time | Structure | *p* | *t* Value | Std. | Percentile 95% CI | |
|---|---|---|---|---|---|---|
| | | | | | Lower | Upper |
| Pre-test | Student participation | 0.817 | 0.770 | 0.095 | −0.117 | 0.264 |
| | Attitude to mathematics | 0.182 | 6.221 | 0.144 | 0.608 | 1.183 |
| | Mathematical self-efficacy | 0.541 | 6.121 | 0.184 | 0.758 | 1.493 |
| | Basic mathematical competencies | 0.112 | 0.798 | 0.178 | −0.213 | 0.496 |
| Post-test | Student participation | 0.037 ** | −28.353 | 0.093 | −2.817 | −2.446 |
| | Attitude to mathematics | 0.000 *** | −13.902 | 0.115 | −1.832 | −1.372 |
| | Self-efficacy | 0.000 *** | −15.284 | 0.113 | −1.946 | −1.496 |
| | Basic mathematical competencies | 0.013 ** | −14.991 | 0.160 | −2.724 | −2.084 |

Note: ** $p < 0.05$, *** $p < 0.01$, same below.

**Table 10.** Comparison between groups of pre and post-tests in the experimental and control classes.

| Form | Testing Time | Structure | R-Square | F | *p* |
|---|---|---|---|---|---|
| Comparison between groups | Pre-test | Student participation | 0.090 | 0.594 | 0.444 |
| | | Attitude to mathematics | 13.436 | 38.704 | 0.059 |
| | | Mathematical self-efficacy | 21.220 | 37.470 | 0.183 |
| | | Basic mathematical competencies | 0.336 | 0.636 | 0.043 ** |
| | Post-test | Student participation | 115.948 | 803.870 | 0.000 *** |
| | | Attitude to mathematics | 42.956 | 193.276 | 0.000 *** |
| | | Mathematical self-efficacy | 49.602 | 233.610 | 0.000 *** |
| | | Basic mathematical competencies | 96.760 | 224.728 | 0.000 *** |

**Table 11.** Comparison of pre-and post-test means between the experimental and control classes.

| Form | The Average Value of Each Configuration | | | | | | | |
|---|---|---|---|---|---|---|---|---|
| | Pre-Test | | | | Post-Test | | | |
| Classes | Student participation | Attitude to mathematics | Mathematical self-efficacy | Basic mathematical competencies | Student participation | Attitude to mathematics | Mathematical self-efficacy | Basic mathematical competencies |
| Control classes | 2.500 | 3.450 | 3.270 | 1.820 | 2.870 | 3.250 | 2.880 | 2.680 |
| Experimental classes | 2.422 | 3.560 | 3.150 | 1.680 | 3.900 | 4.150 | 3.910 | 4.177 |

## 5. Conclusions

### 5.1. Whether the Model Assumptions Are Valid

As can be seen from the collated results (see Table 12), the results of hypothesis 1 of this study are all non-rejected, indicating that hypothesis 1 is valid, meaning that the

structural model has a good fit; the results of hypothesis 2 are rejected, proving that hypothesis 2 is not valid. Indicating that there is a significant difference between the effects of mathematical self-efficacy and basic mathematical competence, and that experiential learning and mathematical attitudes significantly different from mathematical self-efficacy and basic mathematical competence. There is also a significant difference. Hypothesis 3 is not rejected and the Z-values for the indirect, direct, and total effects of experiential learning on mathematical self-efficacy are all significant at greater than 1.96; thus, there is a partial mediating effect of experiential learning in mathematics on mathematical self-efficacy.

**Table 12.** Structural model assumptions and results.

| Hypothetical Content | Results |
| --- | --- |
| Hypothesis 1: $H_0$: The expectation model for experiential learning in mathematics does not differ from the covariance matrix to the sample covariance matrix. | No refusal |
| $H_{01}$: The basic mathematical ability expectation model and covariance matrix do not differ from the sample covariance matrix. | No refusal |
| Hypothesis 2: $H_1$: There is no difference in the impact of mathematical self-efficacy on basic mathematical ability. | Rejection |
| Hypothesis 3: $H_1$: There is a partial mediating effect of experiential learning in mathematics on mathematical self-efficacy. | No refusal |

*5.2. Implications of Structural Models*

1. In this model analysis of experiential learning and basic competencies in mathematics, it can be found that there are significant effects of experiential learning in mathematics, mathematical attitudes, mathematical self-efficacy, and basic competencies in mathematics.

2. From the model analysis, it is clear that effective experiential learning, which increases students' engagement in learning, develops positive attitudes toward mathematics and enhances students' self-efficacy in mathematics, which can influence students' basic competencies in mathematics. In other words, teachers' adoption of experiential learning can foster positive attitudes toward learning, promote the enhancement of students' mathematical self-efficacy, and develop good basic mathematical competencies in mathematics learning.

3. Through this study, the impact of students' behavioral engagement on basic competencies for learning was often overestimated. Therefore, it is important to shift the formality of behavioral engagement in regular teaching and learning, and use different ways to truly engage students' minds and bodies in mathematics learning, so that it will have a multiplier effect on promoting students' mathematics learning and the formation of basic mathematics skills.

*5.3. Conclusions from the Experimental Learning*

1. This study is based on the constructed SEM model for experiential learning research, and the results of the data analysis from IBM SPSS 26.0 show that after one year of study, there is a significant difference in learning participation, mathematical attitude, mathematical self-efficacy, and basic mathematical ability compared to the pre-test, which may be due to the growth and accumulation of knowledge of mathematics after one year of study, resulting in an increase in mathematical competence.

2. The experiential learning promoted changes in learning participation, mathematical attitudes, and mathematical self-efficacy of the experimental class students, which also promoted their desire, interest, and self-efficacy in learning mathematics, which in turn, promoted the increase in their basic mathematical ability.

3. Due to the fact that before the experiential learning, the control class had better mathematics achievement tests in the first semester than the experimental class, the control class performed better than the experimental class in basic mathematics ability, while there

was no significant difference in mathematics attitude, mathematical self-efficacy, and student participation in the pre-test of the two classes. After one year of experiential learning, the experimental and control classes showed significant differences in student engagement, mathematical attitudes, mathematical self-efficacy, and basic mathematical competencies. Therefore, based on the structural model developed, it can be inferred that experiential learning in mathematics can increase students' learning engagement in mathematics learning, which in turn affects mathematical attitudes and enhances mathematical self-efficacy, and may lead to students' increased interest and confidence in mathematics learning, which in turn, affects students' performance in basic mathematical competencies.

## 6. Contribution of This Study

In recent years, there has been an increasing amount of research on the development and validation of SEM models, both in psychology and the social sciences. Most domestic SEM models in China are currently utilized in psychology, and the more pedagogical ones are also used to analyze the mediating role of mathematical attitudes and mathematical self-efficacy and to explore the effects on academic achievement in mathematics. This study explores the SEM model of the relationship between experiential learning and basic mathematics skills in primary school mathematics by collecting various data from home and abroad, and verifies that the model has a good fit and stability. In the analysis of the model, it was found that there were significant effects between experiential learning in mathematics, mathematical attitudes, and mathematical self-efficacy, as well as significant effects between mathematical self-efficacy and basic mathematical competence. This study combines two dimensions of mathematical attitudes and mathematical self-efficacy to complement the previous relationship between mathematical attitudes as a single mediating role and mathematical attainment or mathematical self-efficacy as a single mediating role and mathematical attainment.

In addition, this study also presents theoretical models and conceptual model illustrations in the thesis based on the suggestions of various scholars. Additionally, the sample size, model identification, covariance check, and analysis methods are presented in detail in the statistical analysis, which can provide reference ideas for the later development of the model exploration.

In this study, a one-year experimental study was conducted in the researcher's teaching class and analyzed through pre-test and post-test data collection to demonstrate that experiential learning in mathematics can effectively and positively affect students' engagement, mathematical attitudes, mathematical self-efficacy, and basic mathematical competencies, and also provide a strong basis for model validation. It also provides a reference for the active implementation of experiential learning in elementary school mathematics learning at a later stage.

## 7. Future Research Directions for This Study

The focus of this study is on exploring a structural model based on a linear model that is appropriate for the relationship between experiential learning in mathematics and basic competence in mathematics. However, not all of the psychological factors in this model, other than mathematical attitudes and mathematical self-efficacy, are discussed in terms of whether they can be mediated to influence basic mathematical competence. At the same time, basic competencies in mathematics are content competencies that are directly reflected in test results, whereas the model does not explore whether experiential learning through mathematical attitudes and mathematical self-efficacy also has a good fit with process competencies in mathematics. Therefore, to address the above points, the next study will explore the most effective way to improve the exploration and validation of the model between experiential learning and basic mathematical competencies in primary school mathematics.

Due to the age of the students and the limitations of the Chinese mathematics textbooks, experiential learning was taught and learned in numeracy, space, and statistics

in the classes in this study, and it remains to be seen whether these experiential learning strategies are equally effective in terms of general skills.

**Author Contributions:** Writing—original draft, L.S.; Project administration, L.X. All authors have read and agreed to the published version of the manuscript.

**Funding:** This research received no external funding.

**Data Availability Statement:** The data used to support this study are available from the corresponding author upon request.

**Conflicts of Interest:** The authors declare that they have no conflict of interest.

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
