# Peer review of "An SEM Model of Learning Engagement and Basic Mathematical Competencies Based on Experiential Learning"

_applsci, doi:10.3390/app13063650_

Round 1
Reviewer 1 Report
This study is well structured, however the following aspects could be considered.
1. Title could be more catchy one and could be less than 15 words, rather than construction and validation could be more connected to the influence and impact of the study.
2. The conceptual framework formulated could be expanded with literature. A paragraph could be added to strengthen this model with connections to literature. This would make the methodological aspects clarity and strength.
3. Highlight main outcomes as dot points and key take aways, probably that would lead to suggestions for further study more relevant.
Author Response
Point 1: I changed the title to“A study of basic mathematical competencies based on experiential learning in mathematics”。
Point 2: I searched for literature studies on experiential learning and mathematical attitudes, and experiential learning and mathematical self-efficacy, and added them to 3.3. attitudes to mathematics and 3.4. mathematical self-efficacy to support the theoretical basis of the model construct.
Point 3:I have presented the study conclusions as "three points".
Reviewer 2 Report
An exhaustive review of the citations and bibliographical references should be carried out. They are not in accordance with the norm of the magazine. They appear without numbering, acronyms of the names of the authors and the names appear instead of the surnames. Some years of dating do not match the references.
Table 11 is not named in the text. It is not well understood.
The hypotheses should be listed in another way since they give rise to a mistake in their interpretation. Likewise, the clarity in the conclusions
Author Response
Point 1: I have revised the literature。Like “Stanislas Dehaene,Serge Bossini&Pascal Giraux.(1993).The Mental Representation of Parity and Number Magnitude.Journal of Experimental Psychol093z General,3,V01.122,No.3:371-396.”
I don't quite understand clearly what that sentence “They appear without numbering means,maybe I need to revise the literature like “[10]Stanislas Dehaene,Serge Bossini&Pascal Giraux.(1993).The Mental Representation of Parity and Number Magnitude.Journal of Experimental Psychol093z General,3,V01.122,No.3:371-396.”?
Point 2: Table 11 was named “student participation”,“Attitude to mathematics”,“mathematical Self-efficacy”,“Basic mathematical competencies”
Point 3:I have presented the study conclusions as "three points".
Reviewer 3 Report
I would like to thank the author for this research that aims to It aims to Therefore, this study explores the structural model that fits the relationship between experiential learning in mathematics and basic competencies in mathematics using a linear model.
The research subject is timely. It also fits the aim and scope of the journal.
The literature review section is too long. You also used too much concepts. It is very hard to link between them and test them.
The first part of the research is very complicated and difficult to understand.
In the introduction part you cited two times two different research aims that are again difficult to understand.
In the research method section, you did not specify when the research was conducted, where and how you selected the schools?
In recent research, we mix between the literature review and the hypotheses.
The absence of a clear research aim and clear direction of the research make it difficult to accept. I suggest you to reduce the number of variables and simplify the research aim.

Author Response
Point 1: The literature review has been streamlined.And I searched for literature studies on experiential learning and mathematical attitudes, and experiential learning and mathematical self-efficacy, and added them to 3.3. attitudes to mathematics and 3.4. mathematical self-efficacy to support the theoretical basis of the model construct.Also some literature irrelevant to this article is deleted
Point 2:I have made some changes to the introduction section and I added the research purpose and research in chapter 3“Purpose of the study”.the research method section,and how I selected the schools was in the Chapter 4 “4.5 Method of sampling the study sample”.
Point 3:I have presented the study conclusions as "three points".
Round 2
Reviewer 3 Report
The author made necessary changes.